# Hallmarks of Metabolic Reprogramming and Their Role in Viral Pathogenesis

**DOI:** 10.3390/v14030602

**Published:** 2022-03-14

**Authors:** Charles N. S. Allen, Sterling P. Arjona, Maryline Santerre, Bassel E. Sawaya

**Affiliations:** 1Molecular Studies of Neurodegenerative Diseases Lab, FELS Cancer Institute for Personalized Medicine Institute, Lewis Katz School of Medicine, Temple University, Philadelphia, PA 19140, USA; charlesnallen85@temple.edu (C.N.S.A.); sterling.arjona@temple.edu (S.P.A.); maryline.santerre@temple.edu (M.S.); 2Departments of Neurology, Lewis Katz School of Medicine, Temple University, Philadelphia, PA 19140, USA; 3Department of Cancer and Cell Biology, Lewis Katz School of Medicine, Temple University, Philadelphia, PA 19140, USA; 4Department of Neural Sciences, Lewis Katz School of Medicine, Temple University, Philadelphia, PA 19140, USA

**Keywords:** metabolic reprogramming, Warburg effect, virus, viral replication, glycolysis, glutaminolysis, pentose phosphate pathway, mitochondria, lipid metabolism, amino acid metabolism, biomass, biosynthetic and bioenergetic pathways

## Abstract

Metabolic reprogramming is a hallmark of cancer and has proven to be critical in viral infections. Metabolic reprogramming provides the cell with energy and biomass for large-scale biosynthesis. Based on studies of the cellular changes that contribute to metabolic reprogramming, seven main hallmarks can be identified: (1) increased glycolysis and lactic acid, (2) increased glutaminolysis, (3) increased pentose phosphate pathway, (4) mitochondrial changes, (5) increased lipid metabolism, (6) changes in amino acid metabolism, and (7) changes in other biosynthetic and bioenergetic pathways. Viruses depend on metabolic reprogramming to increase biomass to fuel viral genome replication and production of new virions. Viruses take advantage of the non-metabolic effects of metabolic reprogramming, creating an anti-apoptotic environment and evading the immune system. Other non-metabolic effects can negatively affect cellular function. Understanding the role metabolic reprogramming plays in viral pathogenesis may provide better therapeutic targets for antivirals.

## 1. Introduction

Metabolic reprogramming is a crucial area of study in cancer and is listed as a hallmark of cancer [1]. Dr. Otto Warburg first described metabolic reprogramming while studying the properties of tumorigenic cells. He observed an increase in the metabolism of glucose, known as glycolysis, and a subsequent high concentration of lactic acid [2]. Dr. Warburg noted that this increase in glycolysis was taking place even in the presence of oxygen. Typically, the cell utilizes mitochondrial oxidative phosphorylation (OXPHOS) when oxygen is the only available resort to upregulating glycolysis in a hypoxic environment. For that reason, the finding that cancer cells act as hypoxic cells was surprising. The term for this phenomenon is the Warburg effect and is defined as an increase in glucose uptake and increase in lactate production even in the presence of oxygen [3]. A significant role of the Warburg effect is to provide the rapidly proliferating cancer cells with enough “building blocks” to fuel cellular division. These building blocks are also known as biomass (nucleotides, amino acids, and lipids) and are essential for creating new cells [4]. Biomass is generated in a significant quantity, partly through the Warburg effect. Since its characterization, the Warburg effect remains a part of a broader phenomenon referred to as metabolic reprogramming [5].

In addition to cancer cells, viruses also depend on biomass to fuel viral replication and production of new virions; therefore, they depend on metabolic reprogramming [6,7,8]. Most studies concerning metabolic reprogramming arise from cancer models and cancer samples. Like cancer, metabolic reprogramming contributes to viral infections. Here, we summarized the current literature that defines the hallmarks of metabolic reprogramming and their involvement in viral replication.

## 2. Hallmarks of Metabolic Reprogramming

Metabolic reprogramming is an integral part of both cancer and viral infections, not only to provide the materials for the large-scale biosynthesis for rapid replication and survival but also to provide energy for these processes [9,10]. We classified changes in cellular metabolism that contribute to metabolic reprogramming in six main hallmarks: (1) increased glycolysis and lactic acid, (2) increased pentose phosphate pathway (PPP), (3) increased glutaminolysis, (4) mitochondrial changes, (5) increased lipid metabolism, (6) changes in amino acid metabolism, and (7) changes in other biosynthetic and bioenergetic pathways (Figure 1). These hallmarks play crucial roles in viral pathogenesis. Hence, it is necessary to describe these hallmarks’ involvement before understanding how viruses take advantage of them.

## 3. Glycolysis

An increase in glycolysis, glucose uptake, and lactic acid production was the first hallmark of metabolic reprogramming recognized by Dr. Warburg in 1924; therefore, it is referred to as the Warburg effect [3,11]. In normal cells, glucose is converted into pyruvate through regulated enzymatic reactions called glycolysis. Pyruvate is then transformed into acetyl coenzyme A (acetyl-CoA), which fuels the tricarboxylic acid cycle (TCA cycle), also known as the Krebs cycle [12,13]. Through the TCA cycle, the cell generates nicotinamide adenine dinucleotide (NAD) + hydrogen (H) (NADH) and flavin adenine dinucleotide (FAD) + 2 hydrogens (H2) (FADH2). These products are used in the mitochondrial respiratory chain through oxidative phosphorylation (OXPHOS) to produce 36 molecules of cellular energy in the form of adenosine triphosphate (ATP) [14]. However, during metabolic reprogramming, cells favorably use glycolysis without pyruvate production to produce lactic acid instead, even in the presence of oxygen (Figure 2A). This phenomenon of glycolysis in the presence of oxygen is known as aerobic glycolysis [15,16]. Because glycolysis produces only two molecules of ATP, cells must compensate for the reduction of ATP by upregulating the amount of glucose that undergoes glycolysis [17,18]. Hence, cells first need to increase the uptake of glucose. Many cells achieve this by activating glucose transporters [19,20]. Additionally, metabolic reprogrammed cells exhibit an increase in glycolytic enzymes to promote further glycolysis [4,21]. Cells experiencing metabolic reprogramming utilize HIF-1α and c-Myc transcription factors, master inducers of glycolysis, to upregulate glucose transporters and glycolytic enzymes (Figure 2A) [15].

Increased glucose uptake and decreased pyruvate, available to enter the TCA cycle, activates the glycolytic pathway. In normal cells, the end product of glycolysis is acetyl-CoA, converted from pyruvate, and this acetyl-CoA can then be converted to citrate and continue through the TCA cycle to produce 36 ATP molecules. However, in metabolic reprogramming, the major end product of glycolysis is lactate instead, which does not enter the TCA cycle [22]. The switch between producing acetyl-CoA from pyruvate to lactate contributes to the upregulation of lactate dehydrogenase A (LDHA), and this switch is beneficial since it helps regenerate NADH to continue the increase in glycolysis (Figure 2A) [23,24].

Increased glycolysis results in the accumulation of many biosynthesis precursors. Many glycolytic metabolites are utilized as building blocks in the biosynthesis of needed cellular products. One of these metabolites, glucose-6-phosphate (G6P), is consumed in the PPP, generating precursors for nucleotide synthesis [25]. Another beneficial glycolytic metabolite is dihydroxyacetone phosphate (DHAP), which is utilized in lipid synthesis. Other metabolites are important in amino acid production and macromolecule synthesis [26,27,28]. The importance of these metabolites in other aspects of cellular maintenance makes increasing glycolysis a considerable step in metabolic reprogramming and a therapeutic target for diseases associated with metabolic reprogramming.

## 4. Pentose Phosphate Pathway

The pentose phosphate pathway (PPP) is a metabolic pathway composed of two arms. The oxidative arm generates NADPH and converts G6P into ribulose-5-phosphate to produce nucleotides through nucleic acid synthesis [29]. The other axis of the PPP is the non-oxidative one that generates fructose-6-phosphate (F6P) and glyceraldehyde-3-phosphate (G3P) from G6P [30]. These products can participate in glycolysis and other metabolic pathways (Figure 2B). The first branch is essential for supplying the cells with ribulose-5-phosphate because of the increased demand for nucleotides. Therefore, the generated NADPH is used for glutathione production, which protects the cell from oxidative stress and apoptosis [31]. NADPH is also used by the cell in the biosynthesis of lipids and other macromolecules, and the increase in NADPH contributes to overall metabolic reprogramming [32].

The increase in the PPP associated with metabolic reprogramming is the result of increased expression of enzymes in the PPP [33,34]. One of the essential elevated enzymes during metabolic reprogramming is glucose-6-phosphate dehydrogenase (G6PD), the first enzyme of the PPP, and it aids in converting G6P to ribulose-5-phosphate (Figure 2B) [29]. G6PD’s enzymatic activity is regulated by the tumor suppressor protein, where in normal cells, p53 associates with G6PD and prevents it from forming heterodimers and becoming enzymatically active [35]. In cells where p53 activity is downregulated, such as cancer cells, G6PD is no longer negatively regulated and contributes to metabolic reprogramming (Figure 2B). Transcriptionally, G6PD can be induced through the HIF-1α protein [36,37]. The activity of HIF-1α increases in metabolic reprogrammed cells [38]. Overall, the increase in the PPP is a hallmark of metabolic reprogramming and understanding how it contributes to this phenomenon may help find better therapeutic targets for diseases associated with metabolic reprogramming.

## 5. Glutaminolysis

Glutaminolysis is a process in which multiple enzymes convert glutamine into TCA cycle metabolites by way of α-ketoglutarate [39]. Glutamine is considered the most abundant amino acid found within the body and is necessary for protein, nucleotide, and lipid synthesis [40]. In metabolic reprogrammed cells, since pyruvate is diverted away from entering the TCA cycle, glutamine is utilized to replenish important metabolites from the TCA cycle to synthesize biomass [41]. The use of glutamine as an anaplerotic substrate for the TCA cycle is the continued production of NADH, FADH2, and electrons that are utilized for ATP production through mitochondrial OXPHOS [42,43].

To trigger glutaminolysis, cells rely on the actions of c-Myc to upregulate glutamine transporters and enzymes required for glutaminolysis (Figure 3A) [44,45]. C-Myc protein increases the expression of glutamine transporters alanine-serine-cysteine transporter 2 (ASCT2) and SN2, resulting in increased cellular uptake of glutamine (Figure 3A) [46]. C-Myc also increases glutaminolysis by suppressing miR-23a/b, which negatively regulates the expression of GLS1, the enzyme required for glutaminolysis [47]. The suppression of miR-23a/b allows the expression of GLS1, which induces the amount of glutamate converted from glutamine that can replenish the TCA cycle (Figure 3A). Glutaminolysis is therefore used to assess the metabolic state of the cell. The dependence on glutamine in metabolic reprogrammed cells may be exploited in therapies for diseases associated with metabolic reprogramming.

## 6. Mitochondrial Changes and TCA Cycle Rewiring

Mitochondria are essential components for many metabolic pathways, namely the TCA cycle and the electron chain. They are crucial for maintaining many aspects of cellular homeostasis [48,49]. During metabolic reprogramming, mitochondria change from primarily being used for energy production, by way of the mitochondrial electron transport chain, to playing a role in generating TCA cycle metabolites to be used in the biogenesis and redox balance [50]. This phenomenon was initially proposed in the Warburg effect as being primarily caused by mitochondrial dysfunction but has since been found to be more of a “rewiring” in the TCA cycle [51].

The rewiring of the metabolic pathway within the mitochondria involves many metabolites, such as citrate, which is the first intermediate in the TCA cycle [52]. In metabolic reprogramming, increased exportation of citrate from the mitochondria to the cytosol occurs (Figure 3B). Once out of the mitochondria, the citrate no longer contributes to the TCA cycle; thus, a “rewiring” has occurred. This exportation is achieved by the mitochondrial citrate carrier (CIC), which exchanges citrate for malate. Once exported, the citrate is used for the biosynthesis of fatty acids [53,54]. In addition to fatty acid biosynthesis, cytosolic citrate produces acetyl-CoA and oxaloacetate, which can be converted further into malate and transported back into the mitochondria by the CIC [55,56].

Another TCA cycle intermediate impacted by this rewiring is aspartate, where the production and exportation to the cytosol are altered (Figure 3B) [57]. Aspartate moves to the cytosol in exchange for glutamate and a proton by the mitochondrial aspartate/glutamate carriers (AGC1 and AGC2) [55]. The newly incoming glutamate can then enter the TCA cycle as α-ketoglutarate as an anaplerotic substrate to continue to fuel the TCA in the absence of pyruvate, characteristic of metabolic reprogramming (Figure 3B). The exportation of aspartate is essential for protein, purine, and pyrimidine biosynthesis and is the rate-limiting event in the biosynthesis of these in tumor cells [58]. The transport of aspartate is part of the larger malate/aspartate shuttle (MAS) and is essential to regenerate cytosolic NAD+ for use in glycolysis [59].

Succinate, another TCA cycle intermediate, acts as a signaling messenger linking TCA cycle rewiring and metabolic reprogramming [60]. Succinate is transported out of the inner mitochondria by the dicarboxylic acid carrier to the cytosol by a voltage-dependent anion channel (Figure 3B) [61,62]. Cytosolic succinate can then interact with and inhibit the oxygen-dependent prolyl hydroxylase (PHD) enzyme, which is responsible for targeting HIF-1α for degradation [63]. Therefore, as succinate is transferred out of the TCA cycle, it contributes to the increase in HIF-1α and thus metabolic reprogramming.

Some of these changes seen in TCA cycle rewiring are due to altered gene expression of either transporters or the enzymes that make up the TCA cycle [50]. One regulator of cell metabolism in the context of metabolic reprogramming is nuclear factor kappa B (NF-κB) [64,65]. NF-κB upregulates the expression of the glucose transporter SLC2A3 and enhances the expression of three TCA cycle enzymes, aconitase 2, isocitrate dehydrogenase 3A (IDH3A), and succinyl-CoA ligase (SUCLA2) [66,67]. Through this regulation, NF-κB upregulates glycolysis and contributes to the rewiring of the TCA cycle and metabolic reprogramming (Figure 3B).

Along with NF-κB, HIF-1α plays a role in switching the mitochondria from an energy producer to processing TCA intermediates for biosynthesis. HIF-1α acts as a master regulator of metabolic reprogramming through the upregulation of genes important for glycolysis [68,69]. Along with upregulating the genes for glycolysis, HIF-1α also increases the expression of lactate dehydrogenase A (LDHA), which inhibits pyruvate dehydrogenase (PDH) through phosphorylation [70,71]. The inhibition of PDH drives the conversion of pyruvate to lactate and away from acetyl-CoA, reducing the amount of acetyl-CoA that enters the TCA cycle for energy production (Figure 3B) [72,73]. In these ways, HIF-1α contributes to metabolic reprogramming by regulating gene expression associated with mitochondrial changes and TCA cycle rewiring.

## 7. Lipid Metabolism

The increase in lipid metabolism is another hallmark because lipids are essential building blocks for cancer and viruses to create new organelles, cells, and virions [74,75]. Lipid synthesis is a multi-enzyme process involving many steps, usually starting with acetyl-CoA and ending in fatty acids [76,77]. In metabolic reprogrammed cells, the majority of acetyl-CoA for lipid synthesis comes from citrate produced by the TCA cycle, and that has been transported out of the mitochondria. In the cytosol, citrate can then be converted back into acetyl-CoA and undergo lipid synthesis (Figure 3C) [78].

Cholesterol is another lipid that increases during metabolic reprogramming [79]. The cholesterol synthesis pathway, the mevalonate, is another branch of lipid synthesis that starts with acetyl-CoA and ends with the conversion of lanosterol into cholesterol (Figure 3C) [80,81]. Cholesterol synthesis is a crucial component of membranes and controls the membrane fluidity and formation of lipid rafts [82]. Cholesterol activates the Ras-Raf signaling pathway, which is often utilized by viruses and cancers to manipulate cellular transcription [83,84].

The sterol regulatory element-binding proteins (SREBPs), a family of transcription factors, control many enzymes involved in fatty acid and cholesterol synthesis [78]. Two different SREBPs are involved in this regulation of lipid synthesis; the first, SRBP1, regulates fatty acid synthesis, while the second, SRBP2, regulates cholesterol synthesis [85,86]. The SREBPs are controlled by AMP-activated protein kinase (AMPK) signaling, and in metabolic reprogramming, increased accumulation of ROS can activate AMPK, thus increasing the transcription of SREBPs, resulting in increased lipid and cholesterol synthesis (Figure 3C) [87,88].

## 8. Amino Acid Metabolism

For metabolic reprogrammed cells to produce biomass at a high rate, amino acids need to be either synthesized, used as carbon sources, used as nitrogen sources, or used as electron transporters [89]. Cells can also use branched-chain amino acids (BCAA), leucine, isoleucine, and valine, to fuel the TCA cycle [90]. Of the BCAAs, leucine is elevated with metabolic reprogramming and is converted into acetyl-CoA, and then fed into the TCA cycle (Figure 3D) [91,92]. Other than BCAAs, threonine can also be catabolized by threonine dehydrogenase (TDH), producing glycine and acetyl-CoA, and subsequentially can enter the TCA cycle as well (Figure 3D) [93,94].

In addition to being used as alternative ways to produce acetyl-CoA for use in the TCA cycle, amino acids can also be used as material for biomass and are required for the synthesis of proteins, lipids, and nucleic acids [89]. Glutamine is used in the synthesis of many non-essential amino acids (NEAAs). For example, in the synthesis of asparagine, glutamine supplies the nitrogen to be used in the conversion of aspartic acid to asparagine (Figure 3D) [95]. Glutamine is also converted to glutamate by glutaminase (GLS), and then glutamate can be further converted to other amino acids such as alanine, aspartate, and phosphoserine [39,96].

Arginine is another amino acid used as a precursor for other NEAAs. An increase in arginine promotes proline synthesis and can also be used as another source of glutamate in metabolic reprogrammed cells [97]. Another amino acid metabolism pathway that is important in the biosynthesis of nucleotides is the conversion of serine to glycine through the serine hydroxymethyltransferase (SHMT), which provides pools of one-carbon methyl groups to be used in de novo nucleotide biosynthesis (Figure 3D) [98].

In addition to NEAA synthesis, amino acid metabolism also supplies carbon atoms for lipid biosynthesis and contributes to acetyl-CoA supply [99]. Amino acid metabolism also contributes nitrogen and carbon for nucleic acid synthesis and is required for purine synthesis [100,101]. The essential amino acid tryptophan can be used in NAD de novo synthesis to replenish NAD levels to continue to fuel the increase in metabolic redox reactions (Figure 3D) [102,103]. In all, amino acid metabolism is essential for metabolic reprogramming through many different pathways ranging from alternative energy to providing materials for the synthesis of proteins, lipids, and nucleotides.

## 9. Other Biosynthetic and Bioenergetic Pathways

The fatty acid oxidation (FAO) pathway is another pathway that is promoted during metabolic reprogramming and provides the cell with ATP through fatty acids (FA) being oxidized in the mitochondria or by cytoplasmic lipophagy [104,105]. FAO is where fatty acids are oxidized into acetyl-CoA and produce NADH and FADH2. Then, the acetyl-CoA is imported into the TCA cycle to generate more NADH and FADH2. Afterward, the NADH and FADH2 are used to fuel OXPHOS for ATP production [106]. Metabolic reprogrammed cells use the acetyl-CoA produced by FAO to energize the TCA cycle, generating additional TCA cycle intermediates, such as citrate, succinate, and aspartate [107,108,109,110].

The increase of citrate through the FAO-fueled TCA cycle turns into α-ketoglutarate or pyruvate, and both reactions generate NADPH [111]. NADPH is essential for maintaining redox homeostasis and cell survival and promoting enzymes to sustain large-scale biosynthesis [112]. The balance between NADPH used in lipid synthesis and NADPH produced in FAO is critical in metabolic reprogramming and is controlled by AMPK [113,114].

The use and production of acetyl-CoA are important in many biosynthesis pathways and post-translational modification of the synthesized products [115,116]. Because of this importance, another pathway metabolic reprogrammed cells can take advantage of to replenish the acetyl-CoA is the conversion of acetate into acetyl-CoA by the mitochondrial-localized acetyl-CoA synthetase 1 (ACSS1) [117]. The production of acetyl-CoA from acetate is crucial for metabolic reprogramming to continuously fuel biosynthesis [118].

## 10. Metabolic Reprogramming and Viruses

Although most metabolic research was performed in the cancer field, viruses also take advantage of these pathways to produce the material and environment needed to sustain a large output of virions [9]. The results of metabolic reprogramming in viral infections provide the virus with free nucleotides for viral genome replication, amino acids for virion assembly, and lipids for membrane formation to envelope the virion [119]. That said, viral infections have been shown to alter the hallmark pathways of metabolic reprogramming.

## 11. Viruses and Glycolysis

Increased glycolysis and lactate production are the typical metabolic changes associated with viral infection [120]. Increased glycolysis also correlates with an increase in glucose consumption and lactate production. This was confirmed using primary human foreskin fibroblast (HFF) cells and non-tumorigenic breast epithelial cells infected with the human cytomegalovirus (HCMV) or adenovirus, respectively [121,122,123,124]. A decrease in oxygen consumption used in mitochondrial OXPHOS was demonstrated in adenovirus-infected non-tumorigenic breast epithelial cells [124]. Similar changes were described in human hepatoma (Huh7) cells infected with the hepatitis C virus (HCV) [125].

Further, HCV also increases the expression of glycolytic enzymes in these cells [126]. Increased glycolysis, lactate production, and glucose uptake have also been shown in Influenza A H1N1-infected cells between 8 and 12 h post-infection and HIV-infected CD4+ T-cells, respectively [127,128,129]. An increase in glycolysis has been directly linked to an active replication of the Dengue virus that continues to manifest for at least 48 h post-infection [130].

Using primary neural progenitor cells infected with Enterovirus A71 (EV-A71), targeted metabolomics analysis resulted in increased glycolytic metabolites. These results indicated that EV-A71 causes a metabolic switch to activate glycolysis. It also demonstrated that increased metabolites are associated with other processes involved in metabolic reprogramming, such as increased pentose phosphate pathway, increased gluconeogenesis, and increased glutamate metabolism [131].

Latent viral infection also increases glycolysis, as demonstrated using endothelial cells infected with latent Kaposi’s sarcoma-associated herpesvirus (KSHV) [132]. KSHV infection also promotes glucose uptake by activating the glucose transporter 3 (GLUT3) and the glycolytic enzyme hexokinase 2 (HK2). HK2 is a glycolysis rate-limiting enzyme [133]. Additionally, KSHV infection encodes multiple microRNAs present in latently expressed regions. These viral miRNAs target two glycolysis regulatory genes, EGLN2 and HSPA9. The repression of these two genes induces HIF-1α and glucose transporter 1 (GLUT1) expression, resulting in increased glycolysis and lactate production in endothelial cells where the KSHV miRNAs were overexpressed [134].

Similarly, the Epstein–Barr virus (EBV) increases the expression of HK2 through the induction of the viral protein latent membrane protein 1 (LMP-1), resulting in an increase in glycolysis [135,136]. This increase in glycolysis has also been observed in cells infected with latent EBV. Latent EBV and KSHV induce glycolysis using viral miRNAs or proteins, suggesting that metabolic reprogramming may be active during viral latency [119].

## 12. Viruses and Pentose Phosphate Pathway

As mentioned, viruses promote changes in the host cell that contribute to metabolic reprogramming. These changes help generate nucleic acids and lipids. The large number of generated nucleotides needed for viral genome replication are supplied by the increases in the non-oxidative branch of the pentose phosphate pathway [137,138]. Proteomics studies showed increased expression of two regulators of the non-oxidative PPP branch, the transketolase (TKT) and transaldolase 1 (TALDO1) enzymes in SARS-CoV-2-infected human Caco-2 cells [139]. Cells infected with the influenza virus display upregulation of two enzymes of the PPP, G6PD and 6-phosphogluconate dehydrogenase (6PGD), resulting in an increased conversion of NADPH and nucleotide production [140].

Several enzymes involved in the PPP were also shown to increase in human adenocarcinoma alveolar epithelial cells (A549). An established cell line uses the AdV-5 viral protein, E1A. The E1A protein induces the expression of 6-phosphogluconolactonase (6PGL) and TALDO1 enzymes, both involved in the non-oxidative branch of the PPP [141]. In latent HCV-infected cells, the rate-limiting PPP enzyme, G6PD, is upregulated, as is NADPH production [142]. In addition to NADPH production, there is also an increase in purine synthesis through the PPP in these cells. In HIV latency, an increase in G6PD and NADPH conversion and the shuttling of glycolytic metabolites to the PPP away from pyruvate production were observed [143]. Therefore, many viruses utilize the non-oxidative branch of the PPP to produce nucleotides needed for replication.

## 13. Viruses and Glutaminolysis

Viral infection also increases glutaminolysis, a hallmark of metabolic reprogramming. In support of this observation, HCMV infection augments the uptake of glucose and glutamine [122]. Further, in cells infected with HCMV and labeled with ^13^C-labeled glucose, an increase in the citrate, malate, and α-ketoglutarate expressions were observed. These results indicate an upregulation of glutaminolysis and subsequent anaplerosis of the TCA cycle [123]. HCMV infection also increases the expression of GLS and glutamate dehydrogenase (GDH) enzymes, both involved in the glutaminolysis pathway. These results indicate that HCMV increases the glutaminolysis to supplement the TCA cycle and maintain the TCA cycle intermediates along with ATP production in the mitochondria [144]. Further, the upregulation of glutamine importer ASCT2 and the increased activity of glutaminolysis regulator c-Myc induce the expression level of glutamine in cells infected with HCMV [132].

In another study where ^13^C-labeled glucose was used in conjunction with HSV-1 infected cells, an increase in oxaloacetate produced by the TCA cycle and increased glutamine anaplerosis was detected [145]. HSV-1 infected cells also showed increased uptake and consumption of glutamine as well as decreased viral output in the presence of a pharmacological inhibitor of GLS [146]. The increase in glutamine consumption and the decrease of viral replication associated with GLS inhibitor suggests that HSV-1 uses metabolic reprogramming, specifically glutaminolysis, to produce virions.

Adenovirus infection causes the same metabolic switch of increasing aerobic glycolysis and glutaminolysis. The induction of glutamine consumption in adenovirus type 5 (AdV-5)-infected cells occurs early during the infectious cycle [124]. Like HSV1, AdV-5 infection increases glutamine transporters ASCT2 and LAT1, allowing for increased glutamine uptake [146]. The increase in glutamine consumption in cells infected with AdV-5 infections is made possible via the reductive carboxylation pathway, which results in glutamine conversion to α-ketoglutarate and further conversion to citrate to generate acetyl-CoA [146].

Other viruses also induce glutaminolysis; for instance, rhinovirus increases glutaminolysis reliance [147]. HeLa cells infected with rhinovirus RV-B114, grown in glutamine-free media, show reduced viral replication, indicating the importance of glutaminolysis in viral replication. Huh7.5 cells infected with HCV showed the same reliance on glutamine, and when infected cells were grown in glutamine-free media, viral replication was hindered [148]. Additionally, HCV-infected Huh7.5 cells showed increased glutamine transport and glutaminolysis-associated anaplerosis of amino acid synthesis [148]. Similarly, HIV-1-infected CD4+ T-cells have also been shown to depend on increased glutaminolysis conversion to α-ketoglutarate to fuel the PPP [149]. We concluded that many viruses rely on metabolic reprogramming, specifically glutaminolysis, for their replication and virion production.

## 14. Viruses and Mitochondrial Changes and TCA Cycle Rewiring

Rewiring of the TCA cycle and changes in mitochondrial function are parts of metabolic reprogramming in infected cells and used in response to ROS to fuel fatty acid synthesis or deregulate cellular acetylation [150]. Increased citrate efflux out of the mitochondria is important to the TCA cycle rewiring. In HCMV-infected cells, there is an increase in citrate cataplerosis through activation of the machinery to transport citrate from the mitochondria to the cytosol [151]. Increased levels of citrate and malate were observed in fibroblasts and epithelial cells infected with HCMV or HSV-1. Using fibroblasts and epithelial cells labeled with ^13^C-glucose, HCMV and HSV-1 trigger a large amount of labeled carbon to incorporate into citrate [145]. However, the number of labeled carbons in citrate varies between the two infections, suggesting different pathways of upregulating citrate are used, indicating TCA cycle rewiring is occurring. In HCV-infected Huh7 cells, increased expression of ATP citrate lyase, an enzyme that converts citrate to acetyl-CoA, correlates with more citrate conversion to acetyl-CoA and increased citrate production from the TCA cycle to fuel the conversion to acetyl-CoA [152,153].

In a metabolomic study of the cerebrospinal fluid (CSF) from people with HIV with neurocognitive impairment (NCI), there were higher levels of citrate and succinate than in the CSF from people without HIV [154]. Interestingly, in patients with HIV, the increase in plasma citrate and succinate correlates with the degree of NCI observed, hinting at the importance of these metabolites, and metabolic reprogramming in a broader sense, in NCI [155]. In CSF from HIV patients on cART, there is still an increase in malate and succinate compared to CSF of uninfected patients, suggesting traditional antivirals may not be targeted enough to restore the TCA rewiring that has occurred [156].

Aspartate is another TCA cycle metabolite that is often altered during metabolic reprogramming. In HCMV, there is an increase in asparagine synthetase (ASNS), the enzyme that converts aspartate to asparagine, and it has been found that a knockout of ASNS inhibits HCMV replication [157,158]. In ^13^C-labeled studies using HSV-1-infected fibroblasts, there is increased incorporation of labeled carbons in aspartate compared to non-infected cells, indicating increased aspartate synthesis [145]. As was seen, many viruses rewire the TCA cycle and use overall metabolic reprogramming for viral replication.

Viruses also cause changes in metabolism, mitochondrial shape, and size [159]. The HIV-1 Vpr alters mitochondrial cristae, which results in mitochondrial swelling and irregular shape [160]. A similar effect was observed in cells infected with SARS-CoV-2 [161]. In neural cells (SF268) infected with Enterovirus 71 (EV71), in addition to cristae loss, the mitochondrial inner membrane appeared discontinuous [162].

An additional mitochondrial function disrupted during viral infections is the production of ROS. Viral infections cause increased cellular ROS through mitochondrial deregulation [163]. Viruses depolarize the mitochondrial transmembrane potential (MMP), which contributes to the accumulation of ROS. HBV depolarizes the mitochondrial membrane through its HBx protein that interacts with VDAC [164]. HCV infections also result in depolarization of the MMP; however, HCV achieves this through inhibiting ETC complex I, resulting in inhibition of electron transfer and increased ROS [165]. HIV-1 increases ROS through mitochondrial damage and dysfunction through the Vpr protein [166,167]. The Enterovirus A71 (EV-A71) protein also increases ROS through the electron transport chain and the loss of cristae structure, resulting in reduced electron transfer [162].

Hence, we concluded that viruses change mitochondrial function due to their abilities to reprogram metabolism. These changes rewire the TCA cycle, which aids in viral replication and contributes to cellular dysfunction.

## 15. Viruses and Lipid Metabolism

Lipid synthesis is essential for viral replication since viruses use lipid membranes to enter the host cell membrane. Lipids are also involved in viral protein maturation and viral envelope production [9]. In HCMV-infected primary HFF cells, and during lytic replication, lipid synthesis is induced by the influx of carbons from glycolysis that enter the fatty acid synthesis pathway [123]. Along with the increase in metabolites in the fatty acid synthesis pathway, HCMV also induces the expression of many enzymes involved in this pathway as well, such as ACC1 and SREBPs [151,168,169,170]. HCMV causes an induction of the carbohydrate-response binding element protein (ChREBP). The expression of ChREBP then induces the expression of many proteins responsible for reactions in the fatty acid synthesis pathway, indicating this pathway may be used by the virus for lipid metabolism [171].

In human epithelial tongue cells infected with EBV, there is an induction of the enzyme fatty acid synthase (FASN). FASN is responsible for the conversion of acetyl-CoA to long-chain fatty acids through the direct induction by the viral protein BRLF1 [172,173]. Further, the EBV latent membrane protein 1 (LMP1) increases the fatty acid synthesis and induces FASN, resulting in increased fatty acids and lipid droplets in EBV-negative Burkitt’s lymphoma (BL) cells [174]. The replication of varicella-zoster (VZV) depends on lipid synthesis. However, the addition of an FASN inhibitor alters this replication and slows fatty acid synthesis, suggesting the importance of lipid metabolism in VZV infection [175].

RNA viruses use lipid synthesis to alter the host cytoplasmic environment to make it more conducive to viral replication [176,177]. HCV infections rely heavily on lipid synthesis since lipids play a role in all aspects of the HCV viral life cycle, from cellular entry through replication on lipid rafts to viral assembly on lipid droplets [178,179]. HCV infection also increases FASN expression and induces SREBPs, resulting in increased fatty acid and lipid synthesis [180,181]. Dengue virus as well requires fatty acid synthesis to replicate [182,183]. This observation confirms that FASN is critical for Dengue virus replication and inhibition of FASN, through chemical intervention or siRNA knockdown, results in decreased viral production.

KSHV infection has been found to induce fatty acid synthesis where latently infected endothelial cells upregulate long-chain fatty acids, and there is a significant increase in lipid droplet staining [132]. If FASN or acetyl-CoA carboxylase, another enzyme responsible for fatty acid synthesis, are inhibited in latently infected cells, these cells undergo apoptosis. The addition of FASN or acetyl-CoA carboxylase inhibitors induces cell death by apoptosis, indicating that fatty acid synthesis is necessary for lipid production and as anti-apoptotic signaling in viral infection.

In HIV-infected RH9 cells, many enzymes in the fatty acid synthesis pathway are elevated, as well as the proteins responsible for an increase in low-density lipoproteins, cellular lipid metabolism, and lipid transport [184]. Similarly, HIV-infected 293T cells show an increase in FASN expression. The addition of an FASN inhibitor reduces HIV virion production by 90% compared to noninhibited infection, indicating the importance of fatty acid synthesis in HIV infection [185].

## 16. Viruses and Amino Acids

Amino acid synthesis is another part of metabolic reprogramming essential for viral infections and viral replication [186]. A significant quantity of amino acids is required to fuel protein synthesis and respond to the high demand for viral protein production [187,188]. In HCMV-infected MRC5 cells, it has been shown that amino acids were taken up from the cell culture medium at a much higher rate than in uninfected MRC5 cells [189]. In addition to increased uptake of amino acids, HCMV infection of MRC5 cells resulted in increased synthesis and secretion of amino acids proline and alanine [189]. Telomerase-immortalized microvascular endothelial (TIME) cells infected with KSHV also showed increases in many NEAAs, including arginine and proline, and increased amino acid metabolism pathways, including proline, arginine, alanine, and aspartate [190].

In EV71-infected Vero cells, the levels of threonine, aspartate, alanine, and glycine decreased, while glutamate, tryptophan, tyrosine, phenylalanine, and leucine increased. Interestingly, EV17 replication depends on the glutamate catabolism pathway [191].

In primary HFFs infected with the Vaccinia virus (VACV), the conversion of aspartate to asparagine was seen to be a rate-limiting step in viral replication and limits the synthesis of viral protein production [192]. Studies also demonstrated a decrease in the expression of argininosuccinate synthetase 1 (AS1) in HSV-infected fibroblasts (an enzyme that converts aspartate to arginine). However, AS1 overexpression decreases viral replication in HSV-infected fibroblasts [193]. The decreased conversion of aspartate to arginine allows for an increased concentration of aspartate to fuel the synthesis of nucleotides [194].

Patients infected with HIV have shown an increase in amino acid breakdown products, mainly the breakdown of tryptophan, and an increase in the amino acid phenylalanine in their blood [195]. Further, HIV infections lead to the depletion of tryptophan, resulting in adverse effects on normal T-cell function [196,197]. It has been shown that HIV infection causes more amino acids to be absorbed from the blood of infected patients to fuel protein synthesis, with threonine and methionine being identified as protein synthesis rate-limiting amino acids [198,199].

## 17. Viruses and Other Biosynthetic and Bioenergetic Pathways

Viral infections can affect other pathways involved in metabolic reprogramming as well; for instance, HCV infection increases the mitochondrial fatty acid oxidation enzyme dodecanoyl coenzyme A-delta isomerase (DCI), resulting in increased FAO [200]. In Dengue-infected Huh7 cells, an increase in FAO, through increased β-oxidation, augments the reliance on fatty acids as an energy source [201]. In SARS-CoV-2 infections, a genome assay discovered that many enzymes in the FAO and β-oxidation pathways are deregulated, resulting in changes in fatty acid synthesis and cellular energy production [202]. In HIV-infected patients, deregulated FAO results in increased fatty acids, leading to fatty liver disease [203].

α-Ketoglutarate production from glutamine has also been identified as a key conversion in many viral infections. For instance, cells infected with HCMV show an increase in GDH activity, resulting in an increase in α-ketoglutarate production [144,177]. Similarly, human respiratory syncytial virus (HRSV)-infected cells display a surge in α-ketoglutarate along with the precursors for α-ketoglutarate conversion [204]. Metabolic analysis performed on samples isolated from patients infected with SARS-CoV-2 showed an accumulation of α-ketoglutarate, suggesting that α-ketoglutarate helps replenish NAD levels to boost the glycolysis pathway [205]. In a cohort of patients with HIV undergoing cART, an increase in α-ketoglutarate and other metabolites was observed. These increases correlate with metabolic syndrome seen in these patients [206].

In metabolic reprogrammed cells, the increased production of α-ketoglutarate also corresponds with an increase in NADPH production through the conversion of α-ketoglutarate to succinyl-CoA through the enzyme alpha-ketoglutarate dehydrogenase (α-KGDH) [207,208]. The increase in NADPH then can be used in many biosynthesis pathways as well as ROS production through NADPH oxidase (NOX) enzymes [209,210]. In H1N1-infected human mucoepidermoid pulmonary carcinoma cells (NCI-H292), an increase in NOX family enzymes, mainly NOX2 and NOX4, was observed. This rise in NOX enzymes led to an accumulation in ROS production [211]. HCV-infected Huh7.5 cells also demonstrated an increased expression of NOX4 and the associated increase of ROS through NOX enzymatic reactions [212]. In SARS-CoV-2 infected patients, a surge in NOX2 was associated with the rise of oxidative stress and was predictive of coronary heart disease associated with SARS-CoV-2 infections [213]. The transfection of astrocytes with an HIV-1 gp120 expression plasmid increases NOX2 and NOX4 expression levels and ROS accumulation [214]. HIV infection also increases NOX1, NOX2, and NOX4 expression levels, all contributing to augmenting ROS production [215]. Thus, metabolic reprogramming also promotes ROS production and cellular oxidative stress.

Viral infections also alter the folate-mediated one-carbon metabolism pathway. The one-carbon metabolism pathway is essential for nucleotide and fatty acid synthesis [216]. In addition to biosynthesis, the one-carbon metabolism modifies the epigenetic pathways through DNA and histone methylation using the folate cycle metabolite S-adenosyl methionine (SAM). SAM is the methyl-donating substrate of methyltransferases that methylates DNA and histones [217]. This phenomenon depends on the serine-glycine-one-carbon pathway (SGOC). Indeed, several genes associated with the folate cycle and glycine biosynthesis were upregulated in mice infected with the Influenza A virus [218]. Inhibition of the folate cycle through targeting dihydrofolate reductase (DHFR) reduces the replication of Influenza A, Influenza B, and RSV [219]. The reduction of viral replication is the result of reduced nucleotide production associated with one-carbon metabolism [220]. Further, cells infected with SARS-CoV-2 showed increased metabolites in the methionine cycle. Hence, SARS-CoV-2 is another virus that relies on one-carbon metabolism to fuel the synthesis of viral RNA and proteins [221].

The folate cycle contributes to the overall cellular metabolic reprogramming [222]. One way the folate one-carbon pathway aids in other metabolic reprogramming pathways is through the recycling of NADP to NADPH, used in the pentose phosphate pathway [223]. Finally, the folate cycle also influences other metabolic pathways through the regulation of serine that helps the conversion of glycine from serine [224]. Serine is a key regulator of glycolysis through the regulation of pyruvate kinase enzymatic activity [225].

## 18. Non-Metabolic Effects of Metabolic Reprogramming

Metabolic reprogramming has many non-metabolic effects on the cell. These can include increased inflammation, anti-apoptosis, immune evasion, and the production and accumulation of advanced glycation end products (AGEs) [226]. Some of these effects are essential for viral replication and promotion of tumors, while others are byproducts of the changes that occur with metabolic reprogramming and help modify the cellular environment (Figure 4) [227].

Inflammation produced by metabolic reprogramming can occur from the increased lactic acid output of pro-inflammatory cytokines produced through increased glycolysis and increased ROS (Figure 4) [228,229,230,231,232,233]. The inflammation produced by metabolic reprogramming can help drive molecular changes to promote more metabolic reprogramming and promote angiogenesis to supply the cells with more nutrients to sustain large amounts of biosynthesis [234,235]. Inflammation produced by metabolic reprogramming can also recruit macrophages that respond to the inflammation but also provide the cell with cytokines such as TNF-α and interleukins, which help promote metabolic reprogramming and growth and inhibit cytotoxic T-cells [236,237,238].

Metabolic reprogramming also acts as an anti-apoptotic signal mainly through the induction of glycolysis (Figure 4) [10]. During metabolic reprogramming, the cells evade apoptosis through HIF-1α. The HIF-1α protein induces glycolytic enzymes, resulting in the decrease of apoptotic signals [239,240]. The HIF-1α protein also decreases the apoptotic pathway through its direct interaction and suppression of the pro-apoptotic protein BH3 interacting-domain death agonist (BID) [241,242]. Like HIF-1α, the Akt protein also regulates glycolysis in metabolic reprogrammed cells. The Akt protein inhibits apoptosis through suppression of the p53 upregulated modulator of apoptosis (PUMA) and glycogen synthase kinase 3 (GSK-3) [243]. Several other molecules of glycolysis are also involved in apoptosis resistance. These include phosphorylation of BCL2-associated agonist of cell death (BAD), modification of cytochrome c by glucose, and the increased role NADPH has in the reduction of apoptosis induced by ROS [243,244,245].

Another result of metabolic reprogramming is the ability of the cells to evade cell death associated with the immune response (Figure 4). One way cells evade the immune system is by increasing lactate production from metabolic reprogramming, which can interfere with T-cell metabolism and decrease T-cell response [246,247]. Lactate can also damage dendritic cells and inhibit monocyte migration, further aiding in the evasion of the immune system [248,249,250,251]. In addition to the acidification of the extracellular space, metabolites can also play roles in the evasion of the immune system through the inhibition of T-cells [252]. There are multiple ways metabolic reprogramming metabolites can affect T-cells; one way is through the alterations in differentiation through the effect of kynurenine, a product of tryptophan catabolism [253]. T-cells can also be affected by an enzyme in the tryptophan catabolism pathway, indoleamine-2,3-dioxygenase (IDO), which inhibits T-cell proliferation and induces T-cell apoptosis [254,255].

Metabolic reprogramming can also result in the formation of AGEs (Figure 4). AGE formation can take place through two different pathways, both resulting in highly reactive dicarbonyls. One pathway that results in dicarbonyls is the formation and degradation of Amadori products. The first step in this reaction is the formation of Schiff bases and rearrangement to form an Amadori product [256]. These Amadori products and Schiff bases can undergo further oxidative breakdown in the presence of metal ions, which produce reactive intermediates, dicarbonyls, such as glyoxal (GO) and methylglyoxal (MG) (Figure 5) [257].

While the production of dicarbonyls by way of Amadori products can take hours to days, dicarbonyls can be formed much faster through the spontaneous degradation of glycolytic intermediates [258]. In this process, glycolytic intermediates DHAP and G3P undergo nonenzymatic degradation and form the dicarbonyl MG (Figure 5) [259]. The reactive dicarbonyls, MG, GO, and 3-deoxyglucosone (3-DG), can then form AGEs by nonenzymatically reacting with the free amino group on lysine and arginine residues [257,260,261,262].

AGEs are classified into three groups with unique characteristics: (1) fluorescent crosslinking AGEs, (2) non-fluorescent crosslinking AGEs, and (3) non-crosslinking AGEs (Figure 5) [260]. Crosslinking AGEs result in the linkage of two proteins through the binding of the AGE to lysine or arginine residues on the proteins [240,263,264]. Non-crosslinking AGEs are AGEs that bind to lysine or arginine residues and create protein adducts. Crosslinking AGEs have been shown to affect protein structure and function and result in reduced enzymatic activity, protein aggregation, altered biophysical properties, and changes in protein–protein interactions [265,266,267,268]. Non-crosslinking AGEs have been shown to change receptor ligands, block receptors, block protein cleavage sites, cause misfolding of protein, and inhibit protein degradation [240,264,269,270,271].

The involvement of AGEs in diabetes is well-documented due to the increased glucose uptake [259]. In patients with diabetes, the increase in AGEs has been linked to increased inflammation, cardiovascular failure, decreased renal function, and diabetic nephropathy [272,273,274,275,276]. AGEs have also been shown to play a role in many different cancers [277]. The binding of AGEs to the receptor for advanced glycation end products (RAGE) promotes angiogenesis and inflammation, leading to increased proliferation, migration, and invasion in cancers [278]. In viral infections, it has been shown that AGE binding to RAGE is the leading cause of inflammation and pro-inflammatory cytokine production [279,280].

In addition to diabetes and cancer, there is additional evidence that AGEs play a role in Alzheimer’s and other neurogenerative diseases [281,282]. We demonstrated that the HIV protein gp120 augments AGE production through increased glycolysis and ROS. This links increased AGE concentration to neurodegeneration and HIV-associated neurocognitive disease (HAND) [283]. Increased inflammation associated with AGEs acts through RAGE in the hippocampus of Alzheimer’s patients and promotes the pathogenesis of the disease [284,285]. AGEs may also accelerate beta-amyloid aggregate formation in Alzheimer’s patients [286,287]. Therefore, the ability of AGEs to bind to an array of proteins aids in the progression of many diseases and implicates metabolic reprogramming in disease pathogenesis.

## 19. Conclusions

The hallmarks of metabolic reprogramming represent the cellular processes deregulated in diseases that affect cellular metabolism. These diseases include cancers and viruses that hijack a cell’s energy source and rewire it to produce biomass needed for proliferation and virion production. However, with more research conducted, more metabolic reprogramming hallmarks could be identified.

Many of the hallmarks of metabolic reprogramming outlined here represent ways in which a cell may feed metabolites back into the TCA cycle. As glycolysis is continuing to be promoted and yet the conversion of pyruvate to acetyl-CoA is not occurring, the use of the TCA cycle for energy slows down. However, the TCA cycle is critical for producing metabolites used in the production of biomass. Therefore, viruses may target any number of enzymes or shuttles at any step along the way to ensure the cell is only working to produce more virions. Over time, as viruses continued to benefit from the non-metabolic effects of metabolic reprogramming, they evolved to generate an environment conducive to the replication of virions and immune evasion. Comprehension of the mechanisms behind metabolic reprogramming and how this phenomenon benefits viruses and cancers will help researchers develop better-targeted drugs and therapies.

## Figures and Tables

**Figure 1 viruses-14-00602-f001:**
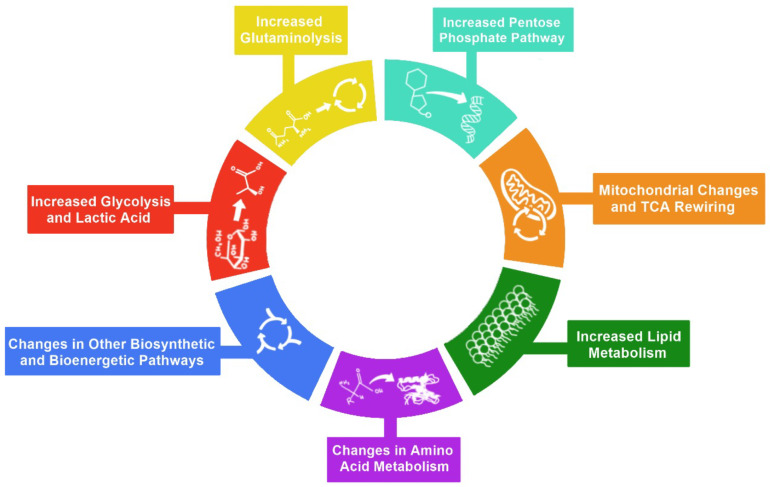
Hallmarks of metabolic reprogramming. The illustration represents the seven hallmarks of metabolic reprogramming: 1—increased glycolysis, 2—increased glutaminolysis, 3—increased pentose phosphate pathway, 4—mitochondrial changes and TCA rewiring, 5—increased lipid metabolism, 6—changes in amino acid metabolism, and 7—changes in other biosynthetic and bioenergetic pathways.

**Figure 2 viruses-14-00602-f002:**
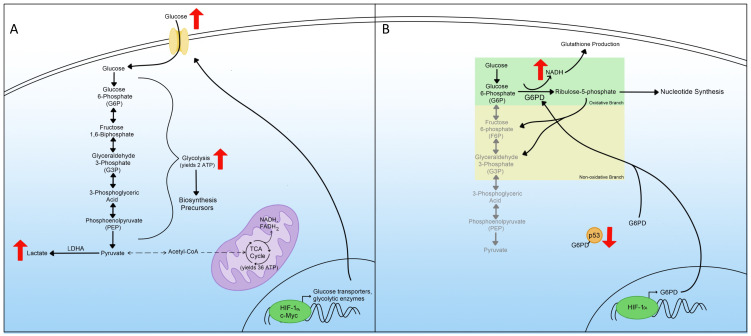
Glycolysis and the pentose phosphate pathway in metabolic reprogramming. (**A**) Upregulated glycolysis in metabolic reprogramming including the transcriptional regulation and increased expression of glucose transporters as well as the consequences of increased glycolysis. (**B**) Mechanisms leading to pentose phosphate pathway (PPP) upregulation during metabolic reprogramming include both branches and transcriptional regulation of critical enzymes.

**Figure 3 viruses-14-00602-f003:**
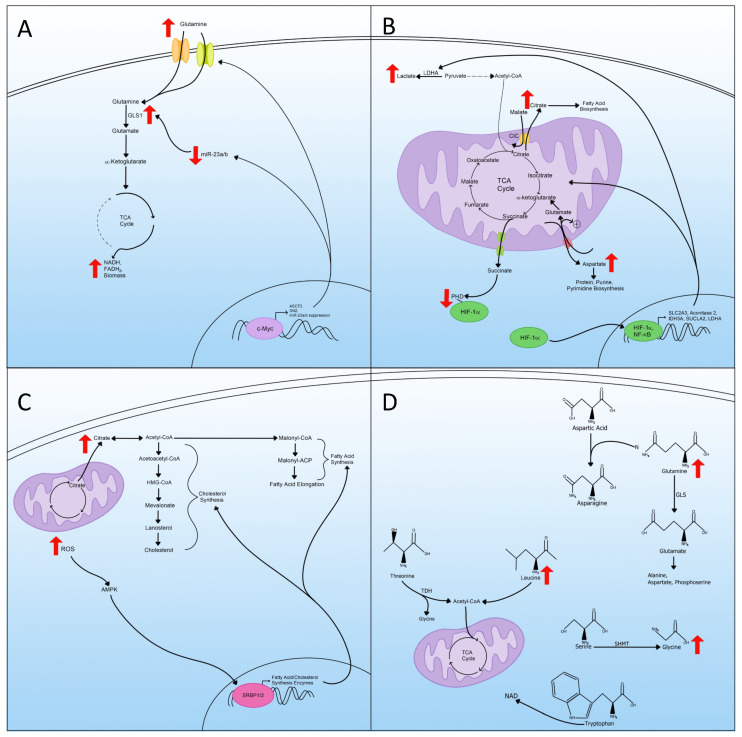
Glutaminolysis, mitochondrial changes, lipid metabolism, and amino acid metabolism in metabolic reprogramming. (**A**) Upregulation of glutaminolysis via transcriptional regulation and the status of miR-23. The cartoon also displays an increased expression of glutamine transporters and enzymes. (**B**) A schematic representation of mitochondrial rewiring of the TCA cycle during metabolic reprogramming includes transporting key TCA metabolites outside the mitochondria and the transcriptional regulation of critical enzymes and transporters. (**C**) Increased lipid metabolism, ROS, and the transportation of citrate out of the mitochondria used for lipid synthesis as a part of metabolic reprogramming. (**D**) Amino acid metabolism regulation in metabolic reprogramming includes the increase in certain amino acids that act as precursors for others.

**Figure 4 viruses-14-00602-f004:**
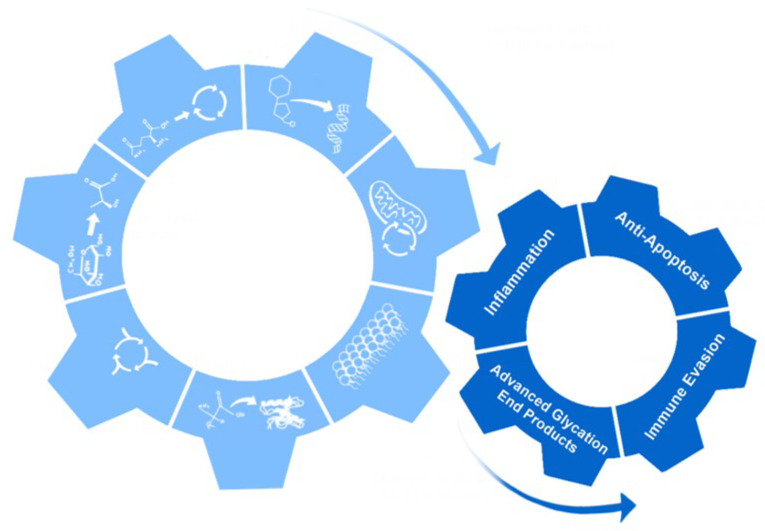
Non-metabolic effects of metabolic reprogramming. The cartoon shows how the seven hallmarks of metabolic reprogramming led to non-metabolic effects: inflammation, anti-apoptosis, immune evasion, and production of advanced glycation end products.

**Figure 5 viruses-14-00602-f005:**
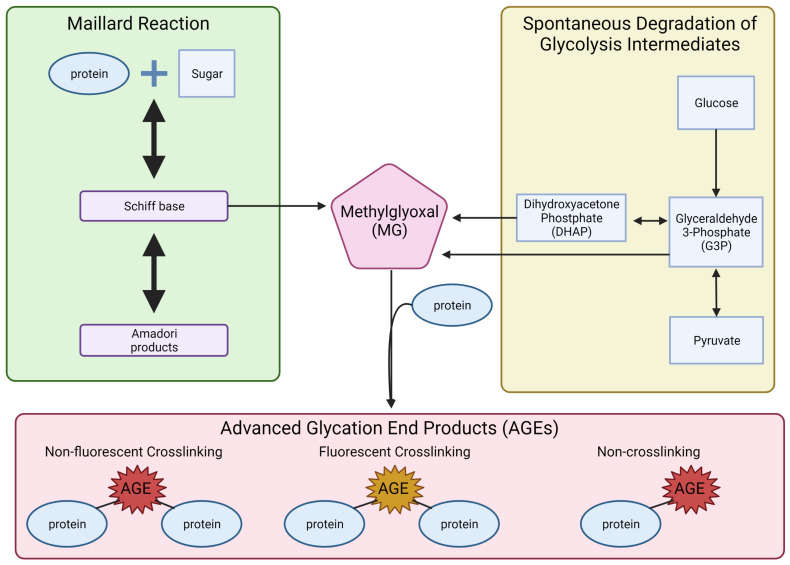
Formation of the advanced glycation end products. The illustration displays two pathways: Maillard reactions and the spontaneous degradation of glycolysis intermediates that lead to the formation of three types of advanced glycation end products.

## Data Availability

Not applicable.

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
