# Peer review of "Hallmarks of Metabolic Reprogramming and Their Role in Viral Pathogenesis"

_viruses, 2022, doi:10.3390/v14030602_

Round 1

Reviewer 1 Report

This review gives an account of the roles of metabolic programming in viral infection. Reprogramming of host cell metabolism is essential to provision of energy and metabolites for viral replication. The altered metabolic pathways include glycolysis, glutaminolysis, pentose phosphate pathway, mitochondrial metabolism, lipid metabolism, amino acid metabolism and energy metabolism. Some mechanistic aspects underlying these changes are also described.

General Comment

  1. Though this review is well-organized, the references included are largely outdated. Of 203 cited references, only 28 references have been published since 2020. Of 28 references, only 7 references are related to virus-induced metabolic changes. Four of them are related to SARS-CoV2 or COVID-19, other are related to HIV and adenovirus. Recently, there has been much progress in this field. For instance, Cheng et al reported the enterovirus-A71 induced metabolic reprogramming is associated with viral protein-host cell protein interaction (Cheng, M.L. et al, Metabolic Reprogramming of Host Cells in Response to Enteroviral Infection. Cells 2020, 9, 473); Zou et al described how EV-A71 virus infection affects glucose homeostasis (Zou, Z. et al, Metabolic Profiling Reveals Significant Perturbations of Intracellular Glucose Homeostasis in Enterovirus-Infected Cells. Metabolites 2020, 10, 302).
  2. It is better to cite more original articles in this review.
  3. As regards the mitochondrial metabolism, the changes in TCA cycle should be discussed in the context of changes to mitochondria themselves (e.g. structural changes, ROS generation, etc). It is exemplified by the virus-induced changes in mitochondrial morphology and structure (Brault C. et al, Hepatitis C Virus-Induced Mitochondrial Dysfunctions. Viruses 2013, 5(3), 954-980; Cheng, M.L. et al, Enterovirus 71 Induces Mitochondrial Reactive Oxygen Species Generation That is Required for Efficient Replication. PLoS ONE 9(11): e113234).

Specific Comment

1. English usage should be brushed up. For example, Line 95: In normal cells, the end-product of glycolysis is acetyl-CoA, converted from pyruvate, which can then be converted to citrate and continue through the TCA cycle to produce 36 ATP molecules. The relative pronoun “which” should have referred to acetyl-CoA. However, in this grammatical structure, “which” refers to pyruvate. The sentences alike should be revised accordingly.

Author Response

We thank the Reviewer for his constructive comments.

1- We did update the list of References and added new ones (~80) and we kept the original ones.

2- We did revise the manuscript and made it more clear.

3- We did add a small paragraph to discuss the changes in the mitochondria and ROS (major changes are marked in Red, see the second red section).

Reviewer 2 Report

The manuscript is a very good survey on metabolic reprogramming and its involvement in virus replicative cycle.

The manuscript is well written and organised. Each section is appropriately described. References are adequate and updated.

  • The manucript, before publication, needs of the following improvements

Include in the paper the folate-mediated one-carbon metabolism pathway (and DHFR as drug target of folate metabolism) which plays an important role in the synthesis of DNA but also in the maintenance of methylation reactions in the cells. This pathway has drawn renewed attention in supporting cancer and virus replication. Describe also the relationship between folate and lipid metabolisms. See the following papers, as an example.

- Folate can promote the methionine-dependent reprogramming of glioblastoma cells Zgheib R, Battaglia-Hsu SF, Hergalant S, Quéré M, Alberto JM, Chéry C, Rouyer P, Gauchotte G, Guéant JL, Namour F. Folate can promote the methionine-dependent reprogramming of glioblastoma cells towards pluripotency. Cell Death Dis. 2019 Aug 8;10(8):596. doi: 10.1038/s41419-019-1836-2.

- Chen L, Zhang Z, Hoshino A, Zheng HD, Morley M, Arany Z, Rabinowitz JD. NADPH production by the oxidative pentose-phosphate pathway supports folate metabolism. Nat Metab. 2019 Mar;1:404-415.

- Tonelli M, Naesens L, Gazzarrini S, Santucci M, Cichero E, Tasso B, Moroni A, Costi MP, Loddo R. Host dihydrofolate reductase (DHFR)-directed cycloguanil analogues endowed with activity against influenza virus and respiratory syncytial virus. Eur J Med Chem. 2017 Jul 28;135:467-478. doi: 10.1016/j.ejmech.2017.04.070.

- Francesconi V, Giovannini L, Santucci M, Cichero E, Costi MP, Naesens L, Giordanetto F, Tonelli M. Synthesis, biological evaluation and molecular modeling of novel azaspiro dihydrotriazines as influenza virus inhibitors targeting the host factor dihydrofolate reductase (DHFR). Eur J Med Chem. 2018 Jul 15;155:229-243. doi: 10.1016/j.ejmech.2018.05.059.

- Zhang Y, Guo R, Kim SH, Shah H, Zhang S, Liang JH, Fang Y, Gentili M, Leary CNO, Elledge SJ, Hung DT, Mootha VK, Gewurz BE. SARS-CoV-2 hijacks folate and one-carbon metabolism for viral replication. Nat Commun. 2021 Mar 15;12(1):1676. doi: 10.1038/s41467-021-21903-z.

I consider this manuscript of interest for readers working in the field, and I suggest its publication after having addressed the following revisions.

Author Response

We thank the Reviewers for pointing at several interesting References that we now included in the revised version.

Also, we added a section regarding the Folate section (see changes in Red, the 4th Red section).

Round 2

Reviewer 1 Report

This review gives an account of the roles of metabolic programming in viral infection. Reprogramming of host cell metabolism is essential to provision of energy and metabolites for viral replication. The altered metabolic pathways include glycolysis, glutaminolysis, pentose phosphate pathway, mitochondrial metabolism, lipid metabolism, amino acid metabolism and energy metabolism. Some mechanistic aspects underlying these changes are also described. 

Comment

Moderate English editing is needed to enhance its readability.